# Heart Failure Alert Duration and Time at Risk of Heart Failure as Potential Modifier Factors of the TriageHF Algorithm in Remote Monitoring of Heart Failure: A Cohort Study

**DOI:** 10.3390/diagnostics15233065

**Published:** 2025-12-01

**Authors:** David Ledesma Oloriz, Daniel García Iglesias, Rodrigo Ariel di Massa, Álvaro Lorente Ros, Fernando López Iglesias, Vanesa Alonso Fernández, José Manuel Rubín López

**Affiliations:** 1Arrhythmia Unit, Cardiology Department, Hospital Universitario Central de Asturias, Avda de Roma s/n, 33011 Oviedo, Spain; 2Instituto de Investigación Sanitaria del Principado de Asturias (ISPA), 33011 Oviedo, Spain; 3Heart Failure Unit, Cardiology Department, Hospital Universitario Central de Asturias, Avda de Roma s/n, 33011 Oviedo, Spain

**Keywords:** TriageHf, heart failure, remote monitoring

## Abstract

**Background**: The TriageHF algorithm provides remote monitoring for heart failure (HF), but its clinical implementation is limited by a high rate of false positive alerts. **Objective**: To assess whether incorporating alarm duration and individual time at risk can improve the diagnostic performance of the TriageHF algorithm. **Methods**: This was a single-center prospective cohort study in which 37 patients with Medtronic ICDs implanted between January 2020 and June 2022 were enrolled. HF events were defined as episodes requiring intravenous diuretics or hospitalization. A total of 609 TriageHF alerts were analyzed. Two strategies were analyzed: a standard approach (high-risk and moderate-risk alerts > 7 days considered positive) and a modified approach (high-risk and moderate-risk alerts > 15 days considered positive). The relationship between time spent in a low-risk state and the algorithm’s positive predictive value (PPV) was also assessed. **Results**: In the standard configuration, sensitivity was 96.7% and specificity was 76.4%, with 81.9% of false positives. The modified approach showed improved specificity (85.6%) and PPV (24.3%), with minimal impact on sensitivity (87%) and negative predictive value (99.1%). There was a significant inverse correlation between time spent at low-risk and individual PPV (R^2^ = 0.64, *p* = 0.018). **Conclusions**: Using a ≥15-day threshold improved the specificity and PPV of the TriageHF algorithm. Incorporating individual time at risk may further refine risk stratification and enhance cost-effectiveness in HF remote monitoring strategies.

## 1. Introduction

Despite the development of new therapeutic approaches in the treatment of patients with heart failure (HF), these patients still present high morbidity and mortality rates [1]. This syndrome intercalates periods of clinical stability with decompensation, which often requires hospital admission for intensive treatment. Furthermore, presenting a hospital admission due to HF decompensation increases not only the incidence of readmission, but also the mortality of these patients in the following months [1,2,3,4,5].

Implantable defibrillators devices (ICDs) are indicated in the treatment of patients with HF and left ventricular dysfunction to reduce their mortality [2,6,7,8]. These devices, besides their capability of detecting and treating ventricular arrythmias, can measure different variables related to HF that can be used for remote monitoring [9,10,11,12,13]. Different algorithms have been developed to generate an estimation of the individual risk of a patient to present with a heart failure (HF) episode in the following days [14,15,16,17,18].

The TriageHF algorithm [16] is one of the algorithms developed for the prediction of HF decompensation risk. This algorithm, by measuring and then combining a series of variables, delivers a daily estimated risk of undergoing a heart failure event in the next 30 days. There are seven variables, including the fluid index measured by Opti VOL [11], the physical activity index, the nocturnal heart rate [9], the variability of the heart rate [19], the burden of atrial fibrillation [20], the percentage of resynchronized pacing, and the detection and treatment of ventricular arrythmias.

The TriageHF algorithm computes an estimated risk in a qualitative manner, showing three different levels of HF decompensation risk within the next 30 days: low, moderate, or high. Previous studies have shown that a low-risk alert was associated with a 0.6% risk, a moderate-risk alert was associated with a 1.3% risk, and finally a high-risk alert was associated with a 6.8% risk of hospital admission in the following 30 days after the alert onset [16]. Later studies have tested this algorithm in different scenarios, showing variable results in terms of specificity and sensibility but with high negative predictive value [21,22]. In general, moderate-risk alerts have better sensibility with low specificity and higher rates of false negatives; meanwhile, high-risk alerts have better specificity but with lower sensibility and more false positives. Last trials [21,23] incorporate a direct phone call, which improves diagnostic accuracy and allows us to perform a direct therapeutic intervention for the patient at risk of heart failure.

The most recent studies have also proven a positive relationship between the time spent in high risk, episodes of heart failure, and general mortality [24]. There was also a recent trial, performed with another ICD algorithm [25], in which only risk alerts lasting more than 7 days were considered positive for the analysis. This suggests that alert duration could be an important characteristic to consider when trying to improve algorithms specificity. There was another trial with this other algorithm [26] where they found that the quantitative value of the alarm was related to its diagnostic capability. They reported higher quantitative values with true positive alerts with a statistically significant lower value in the false positive alerts.

When using these algorithms for the early detection of HF decompensation as a telemonitoring strategy, it is essential to achieve a good balance between sensibility and specificity, given the high workload that false positive alarms cause. In this context, although home monitoring has shown clear benefits in the management of patients with implantable devices [23,27,28], the high rate of false alerts has been reported to significantly increase maintenance costs, which may limit their widespread implementation.

Regarding these last findings, the duration of the alerts and the time spent at risk could be potential tools to optimize the diagnostic features of the TriageHF algorithm [29].

## 2. Materials and Methods

### 2.1. Hypothesis

Moderate-risk alerts in the TriageHF algorithm could be redefined to enhance the algorithm’s performance in predicting HF decompensation. In this regard, the duration of alerts may play a crucial role. Therefore, disregarding short-duration alerts may improve the diagnostic accuracy of the TriageHF algorithm without affecting its negative predictive value.

### 2.2. Objectives

To perform a first analysis, in which moderate-risk alerts where the duration was less than 7 days were not considered, and use these results to make a further comparison varying the threshold.To test if considering positive alerts as only moderate-risk alerts where the length was longer than 15 days results in better specificity.To test if there are differences in the characteristics of the sample when dividing them into three groups according to their level of heart failure risk during the follow-up.To test if there are differences in the basal characteristics and the clinical follow-up of the sample depending on the occurrence of a heart failure episode.To analyze if there is a relationship between the time spent in low risk and the positive predictive value of the TriageHF alerts.

### 2.3. Study Population

This is a descriptive, single-center cohort study including a series of patients who received a Medtronic implantable cardioverter defibrillator (ICD) between January 2020 and June 2022. The ICD could be a single-chamber ICD or a CRT-ICD, and it must have the TriageHF feature activated. The baseline clinical characteristics were recorded at the time of implantation.

All patients underwent in-person follow-up visits one month after implantation and at least once annually thereafter. In addition, continuous remote monitoring was conducted using the Medtronic Care link^TM^ home monitoring system (version 3.2.2). At the end of the follow-up period, all patients were contacted by telephone, and their electronic medical records were reviewed to identify any undetected heart failure event. Follow-up ended in January 2023.

This study was approved by the Ethics Committee of the Principality of Asturias (CEImPA 2024.488), and informed consent was obtained from all participants.

### 2.4. Patient Follow-Up

An HF event was defined as a clinical episode of congestion requiring intravenous loop diuretic therapy in the emergency department or hospital admission. Every month in which a patient remained free of HF decompensation was categorized as a negative case of HF.

For the general analysis of alerts, we defined a positive alert as the algorithm’s moderate-risk alert lasting more than 7 days or a high-risk alarm; moreover, moderate-risk alerts of less than 7 days were categorized as low-risk alerts. We performed a second analysis in which we established a threshold of 15 days for considering moderate-risk alerts as positive alarms (those that lasted less than 15 days were considered negative alarms). In both analyses, we considered all high-risk alerts (independent of their duration) as positive HF risk alerts. All low-risk alerts were considered negative alarms for both analyses.

Finally, the total amount of time each patient spent on positive HF risk alert during follow-up (considering only the high-risk alerts and the moderate-risk alerts lasting more than 7 days) was recorded. Based on this, we analyzed the relationship between the proportion of time spent in the low-risk state and the positive predictive value (PPV) of the heart failure alerts. In this analysis, each patient’s PPV was used as the dependent variable, and the time spent in the low-risk state was used as the independent variable.

### 2.5. Statistical Analysis

Continuous variables were expressed as mean ± standard deviation, and categorical variables as absolute frequency and percentage. The comparison of continuous variables between patients with and without clinical events or with and without HF risk alerts was performed using Student’s *t*-test, following confirmation of normal distribution. For categorical variables, the Chi-square test was used, with Yates’ correction applied when appropriate. All confidence intervals were reported at the 95% level.

The diagnostic performance of the TriageHF algorithm was assessed by calculating its sensitivity, specificity, PPV, and negative predictive value (NPV) using clinical HF events identified during follow-up as the reference standard.

To assess the relationship between time spent on negative HF risk alert state and the PPV of alerts, a Pearson correlation test was performed.

Two stratified analyses based on the patient profile were performed. In the first one, the population was divided into three different categories based on the patients’ TriageHf risk alarms: low-risk alerts (patients who spent 100% of the time at low risk, including moderate-risk alerts lasting less than 7 days), moderate-risk alerts (if they had only moderate-risk alerts during follow-up), and high-risk alerts (if they had at least one high-risk alarm during follow-up). For the second one, the population was divided into two groups depending on whether they presented a heart failure event during follow-up. All statistical analyses were performed using R software version 4.5.0 [30].

## 3. Results

### 3.1. Population Characteristics

A total of 37 patients met the inclusion criteria with a mean follow-up time of 494.8 (+/−276.6) days. A total of 81% of the patients were male, with a mean age of 63.86 years. A total of 83% of the patients received CRT-ICD and 17% a single chamber ICD. Almost 70% of the population had a severely reduced left ejection fraction (<30%), with 40% of ischemic etiology. A total of 80% of the patients were previously treated with betablockers, 85% with ARNI or an ACE-I/ARB-II, 49% with MRA, and only 28% had previous SGLT2-I. The basal characteristics of the sample are shown in Table 1.

During the study period, a total of 609 alerts were analyzed, and 31 HF events were recorded. A total of 14 out of the 37 patients experienced an HF episode (37.83%). This traduces an HF incidence of 0.05 episodes per month. A total of 166 HF alerts were generated during the follow-up period, resulting in an alert incidence of 0.27 HF risk alerts per month.

### 3.2. Alert Analysis

A total of 609 alerts were analyzed, wherein 166 of these alerts were positive HF risk alerts and 443 were negative alerts. The sensibility was 96% and the specificity was 76.4%, with 136 (81.9%) false positive alerts and 1 (0.2%) false negative alert. Patients spent 80.4% (±20.3%) of the time at low risk (with an average of 13 months/patient of time at low risk), 17.34% (±17.6%) at moderate risk, and 1.78% (±4.68%) at high risk.

### 3.3. Prespecified 15-Day Cutoff Analysis

When considering positive HF risk alerts as the moderate-risk alerts lasting more than 15 days and all of the high-risk alerts, we reported a total of 111 positive HF risk alerts, with 84 (75%) false positive alerts and 4 (0.08%) false negative alerts. In this analysis, we obtained a sensibility of 87% with a specificity of 85%. Table 2 shows a comparison between the default and the 15-day threshold strategies. This strategy resulted in 55 positive alarms less than the standard strategy, with 53 (96%) of them turning out to be false positive alarms.

### 3.4. Population Follow-Up

When comparing the basal characteristics of the sample based on the risk profile, we observed some relevant differences. We defined three different groups, as follows: the first one included patients who only had low-risk alarms, the second one included those with only moderate-risk alarms, and the third one included those with at least one high-risk alarm during their follow-up. We found some differences between the groups, the most relevant ones being the age of the patients, the prevalence of chronic kidney disease, the occurrence of a heart failure event, and the positive predictive value (PPV). These differences are shown in Table 3 and in Figure 1.

We also performed a comparative analysis depending on the occurrence of a heart failure episode during the follow-up. When comparing these two populations, we observed higher PPV and less time at low risk in the heart failure event group. These differences are shown in Table 4.

### 3.5. Relationship Between Time at Risk of Heart Failure and Positive Predictive Value

Our regression test showed a significant inverse relationship between the amount of time spent in a low HF risk state and the PPV of the HF alerts, with higher PPV values in patients who had been at risk of an HF episode for a longer period of time. The correlation test was significative, with a R^2^ value of 0.64 (*p* = 0.018). Figure 2 shows this analysis.

## 4. Discussion

Many trials have been previously designed in the field of telemonitoring with both TriageHF and HeartLogic algorithms. These trials have tested different strategies trying to find one that is not only precise, but also cost-efficient. Our study shows similar findings to those that have been previously reported. The TriageHF algorithm has a good negative predictive value but with the drawback of high rates of false positive alerts [16,24,31]. This disadvantage is closely related to the cost–benefit of using this algorithm in daily clinical practice. This study was designed to explore potential ways to optimize this remote monitoring strategy.

We explored two potential strategies for this purpose. The first one was testing the algorithm diagnostic features using a modified threshold aimed at reducing the number of positive alarms needed to evaluate and the other was the relation between the time at risk of each patient and the positive predictive value of their own heart failure risk alerts.

Regarding the first strategy, moderate-risk alerts were analyzed with two different protocols. In the first protocol, only alerts lasting more than 7 days were considered positive; moreover, when comparing our results with those of previous trials, we observed that this strategy did not result in an important increase in false negative alerts, showing similar sensibility and specificity to those of other, previous trials. With the second protocol, we only considered positive alerts as those lasting more than 15 days. We found that ignoring moderate-risk alerts lasting less than 15 days resulted in 55 less positive alarms; moreover, 96% of them were false positive, with increased specificity and positive predictive value, decreased sensibility, and almost the same negative predictive value. This is a very interesting finding in the applicability of the algorithm in real clinical practice.

Regarding the second strategy, we found that the longer a patient was out of the low-risk HF state, the higher positive predictive value their own heart failure alarms had, with an inverse moderate correlation between PPV and % of time at low risk of heart failure.

When comparing the groups based on the occurrence of a heart failure event (Table 4), we found a difference in the amount of time at heart failure risk and an almost significant *p* value in the chronic prescription of diuretics. When we analyzed the difference between the three prespecified risk groups, we found that patients who had only low-risk alarms were younger, with less chronic kidney disease and loop diuretic prescription, but with higher prevalence of SGLT2-I use (Table 3).

Our findings suggest that an interventional strategy based on the TriageHF algorithm in which only alerts lasting more than 15 days are considered positive would be safe and feasible. Moreover, we could incorporate the time spent at low risk into this strategy as another variable, which could help to select patients with higher positive predictive value, making the algorithm not only more precise, but also more cost-effective. This could be integrated into a hybrid approach that combines algorithm monitoring with evaluation phone calls. This perspective aligns with emerging evidence from home-based and telemonitoring approaches for managing heart failure and arrhythmias in patients with cardiac implantable devices, as highlighted by Matteucci et al. [31].

By using our 15-day approach, we would reduce the total number of heart failure alerts needed to review, without a significative impact on the sensibility and negative predictive value of the algorithm, reducing the number of phone calls needed to perform in case of incorporating this strategy into a hybrid protocol. According to our findings, this could be even more suitable for patients who spend more time at low risk of heart failure during the follow-up. Based on our results, we believe it would be worthwhile to assess whether individualized strategies, tailored to patients’ clinical characteristics and their pre-test probability of experiencing a heart failure event, would lead to improved cost-efficiency of TriageHF telemonitoring.

One potential limitation to our findings is that, in our trial, episodes of heart failure were defined based on a retrospective revision, which could theoretically lead into undiagnosed heart failure events. This could potentially overestimate the negative predictive value and the specificity of our study. Another important potential limitation is the small sample size, which may affect the external validity of the results. However, despite the limited number of heart failure events and the sample size, over 600 alarms were analyzed, providing substantial data for evaluation.

## 5. Conclusions

According to our results, defining moderate-risk alerts with a threshold of 15 days would increase the specificity and reduce the number of false positive alerts of the TriageHF algorithm without producing a significant effect on the negative predictive value. Also, the total amount of time at risk of heart failure, represented by a TriageHF alarm, seems to correlate with a better positive predictive value of the heart failure alarms. A more cost-efficient strategy based on the TriageHF algorithm could be designed combining this 15-day threshold with the amount of time spent at low risk for each patient.

## Figures and Tables

**Figure 1 diagnostics-15-03065-f001:**
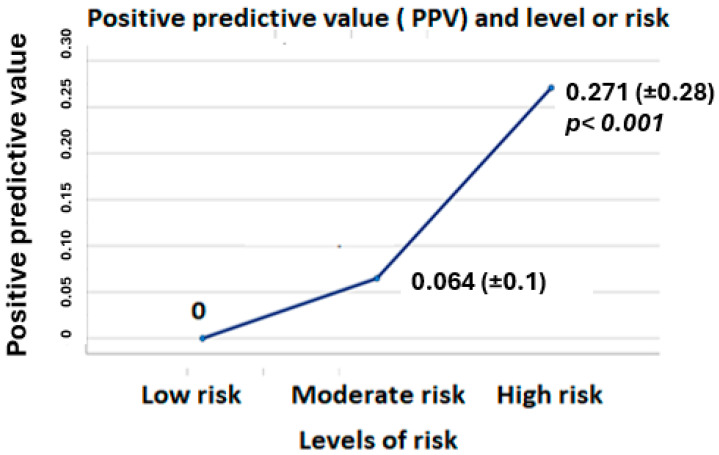
Differences in PPV between the three different groups of risk.

**Figure 2 diagnostics-15-03065-f002:**
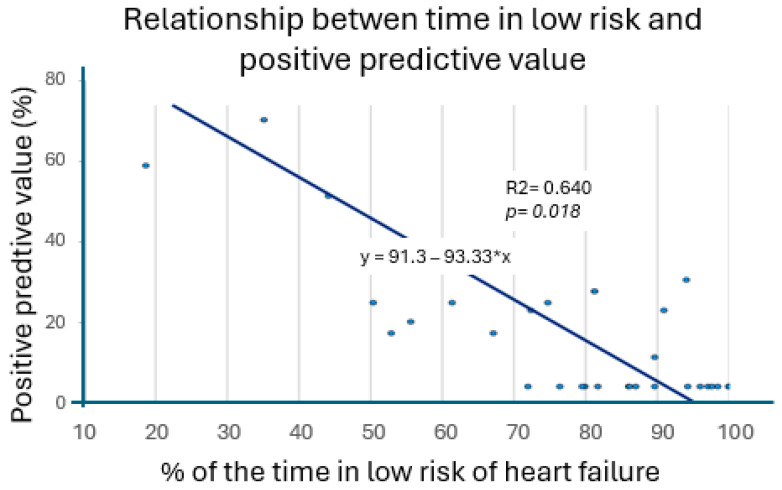
Relation between positive predictive value and percentage of time at low risk.

**Table 1 diagnostics-15-03065-t001:** Basal characteristics of the sample.

VARIABLE	TRIAGE-HF (*N* = 37)
**Male sex**	30 (81%)
**Mean age at implantation**	63.86 (+/−11.407)
**CRT-ICD**	31 (83%)
**Ischemic cardiomyopathy**	16 (43%)
**Primary prevention implantation**	29 (78%)
**Arterial hypertension**	24 (65%)
**Diabetes mellitus**	12 (32.4%)
**Chronic kidney disease**	7 (19%)
**Dyslipidemia**	19 (51.3%)
**Previous smoking**	21 (58%)
**Previous COPD**	5 (13.5%)
**Atrial fibrillation**	10 (27%)
Paroxistic/persistent atrial fibrillation	2 (5.4%)
Permanent atrial fibrillation	8 (21.6%)
**Previous left ventricle ejection fraction (LVEF)**	
>53%	4 (10.8%)
40–50%	1 (2.7%)
30–40%	6 (16.21%)
<30%	26 (70.27%)
**Previous treatment**	
Betablockers	30 (81%)
ACE-I/ARA-II	16 (43%)
MRA	18 (49%)
ARNI	14 (37.8%)
SGLT2-I	11 (28%)
**Previous NYHA functional class**	
I	2 (5.4%)
II	27 (72.9%)
III	6 (16.2%)
IV	2 (5.4%)
**Mean follow-up (days +/− SD)**	494.8 (+/−276.6)
**Confirmed positive cases**	31
**Clinical HF episodes/patient (mean +/− SD)**	0.837 (+/−1.726)

ACE-I: Angiotensin-Converting Enzyme Inhibitor; ARA-II: Angiotensin Receptor Antagonist; ARNI: Angiotensin Receptor–Neprilysin Inhibitor; COPD: Chronic Obstructive Pulmonary Disease; CRT-ICD: Cardiac Resynchronization Therapy with Implantable Cardioverter Defibrillator; LVEF: Left Ventricular Ejection Fraction; MRA: Mineralocorticoid Receptor Antagonist; NYHA: New York Heart Association; SGLT2-I: Sodium–Glucose Cotransporter 2 Inhibitor.

**Table 2 diagnostics-15-03065-t002:** Diagnostic features when considering a 7- and 15-day threshold for moderate-risk alerts.

	Default Configuration	Positive If 15 Days or More in Moderate Risk
**Total number of alerts**	609	609
**Positive cases**	31	31
**Negative cases**	578	578
**Positive alerts**	166	111
**Negative alerts**	443	498
**True positive alerts**	30	27
**False positive alerts**	136	84
**True negative alerts**	442	494
**False negative alerts**	1	4
**Sensibility**	96.7%	87%
**Specificity**	76.4%	85.6%
**Positive predictive value**	18%	24.3%
**Negative positive value**	99.7%	99.1%

**Table 3 diagnostics-15-03065-t003:** Comparison between different risk levels.

VARIABLE	ONLY LOW RISK (*N* = 8)	ONLY MODERATE RISK (*N* = 18)	HIGH RISK (*N* = 11)	*p*
**Male sex**	5 (62.5%)	16 (88.8%)	9 (81.81%)	0.284
**Mean age at implantation**	58.12 +/− 4.09	64.278 +/− 2.72	67.36 +/− 3.48	<0.001
**CRT-ICD**	7 (87.5%)	15 (83.3%)	9 (81.81%)	0.944
**Ischemic cardiomyopathy**	2 (25%)	8 (44.4%)	6 (54.54%)	0.434
**Primary prevention implantation**	7 (87.5%)	12 (66.6%)	10 (90.9%)	0.238
**Arterial hypertension**	5 (62.5%)	11 (61%)	8 (72.7%)	0.807
**Diabetes mellitus**	1 (12.5%)	8 (44.4%)	3 (27.27%)	0.250
**Chronic kidney disease**	0 (0%)	3 (16.6%)	5 (45%)	**0.046 ***
**Previous COPD**	0 (0%)	4 (22%)	1 (9%)	0.272
**Atrial fibrillation**	2 (25%)	4 (22%)	4 (36%)	0.790
Paroxysmic/persistent AF	0 (0%)	1 (5%)	1 (9%)	
Permanent atrial fibrillation	2 (25%)	3 (16.6%)	3 (27.27%)	
**Previous LVEF**				0.367
>53%	1 (12.5%)	3 (16.6%)	0 (0%)	
40–50%	1 (12.5%)	0 (0%)	0 (0%)	
30–40%	1 (12.5%)	4 (22.2%)	1 (9%)	
<30%	5 (62.5%)	11 (61.2%)	10 (91%)	
**Previous treatment**				
Betablockers	7 (87.5%)	13 (72.2%)	10 (91%)	0.560
ACE-I/ARA-II	3 (37.5%)	7 (38.38%)	6 (54.54%)	0.684
MRA	5 (62.5%)	7 (38.38%)	6 (54.54%)	0.247
ARNI	4 (50%)	5 (27.77%)	5 (45.45%)	0.537
SGLT2-I	5 (62.5%)	5 (27.77%)	1 (9%)	0.69
**Chronic diuretic treatment**	5 (62.5%)	7 (38.38%)	11 (100%)	**0.028 ***
**Islgt2 during follow-up**	8 (100%)	11 (61.11%)	5 (45.45%)	**0.044 ***
**Previous NYHA**				0.905
I	1 (12.5%)	1 (5.5%)	0 (0%)	
II	6 (75%)	13 (72.2%)	8 (72.7%)	
III	1 (12.5%)	3 (16.6%)	2 (18.18%)	
IV	0 (0%)	1 (5.5%)	1 (9.09%)	
**Mean follow-up (days +/− SD)**	349.3 +/− 99.1	512 +/− 66.09	575 +/− 84.053	**0.001 ***
**Confirmed positive cases**	0 (0%)	7 (22.58%)	24 (77.41%)	**0.04 ***
**Clinical HF episodes/patient (mean +/− SD)**	0 (0%)	0.388	2.18	
**% of time at low risk**	100 (+/−0)	84.705 (+/−12.92)	60.59 (+/−21.1)	**<0.001 ***
**PPV**	0	0.064 (+/−0.1)	0.271 (+/−0.28)	**<0.001 ***

ACE-I: Angiotensin-Converting Enzyme Inhibitor; ARA-II: Angiotensin Receptor Antagonist; ARNI: Angiotensin Receptor–Neprilysin Inhibitor; COPD: Chronic Obstructive Pulmonary Disease; CRT-ICD: Cardiac Resynchronization Therapy with Implantable Cardioverter Defibrillator; LVEF: Left Ventricular Ejection Fraction; MRA: Mineralocorticoid Receptor Antagonist; NYHA: New York Heart Association; PPV: Positive predictive value; SGLT2-I: Sodium–Glucose Cotransporter 2 Inhibitor. Percentages represent the number of cases in each category in each of the three groups. * *p* value < 0.05.

**Table 4 diagnostics-15-03065-t004:** Comparison based on the occurrence of heart failure decompensation.

VARIABLE	HF EPISODE (*N* = 14)	NOT IC EPISODE (*N* = 23)	*p*
**Male sex**	12 (85.71%)	18 (78.2%)	0.459
**Mean age at implantation**	66.57 (+/−9.7)	62.22 (+/−12.24)	0.24
**CRT-ICD**	12 (85.71%)	19 (82.6%)	0.593
**Ischemic cardiomyopathy**	7 (50%)	9 (39.13%)	0.379
**Primary prevention implantation**	12 (85.71%)	17 (73.9%)	0.340
**Arterial hypertension**	10 (71.42%)	14 (60.8%)	0.387
**Diabetes mellitus**	5 (35.71%)	7 (30.43%)	0.507
**Chronic kidney disease**	3 (21.42%)	5 (21.73%)	0.215
**Previous COPD**	1 (7.14%)	4 (22.22%)	0.63
**Atrial fibrillation**			0.935
Paroxysmic/persistent atrial fibrillation	1 (7.14%)	1 (4.34%)	
Permanent atrial fibrillation	3 (21.42%)	5 (21.73%)	
**Previous LVEF**			
>53%	1 (7.14%)	3 (13.04%)	0.5
40–50%	0 (0%)	1 (4.34%)	
30–40%	1 (7.14%)	5 (21.74%)	
<30%	12 (85.71%)	14 (60.87%)	
**Previous treatment**			
Betablockers	13 (92.85%)	17 (73.9%)	0.122
ACE-I/ARA-II	8 (57.14%)	8 (34.78%)	0.175
MRA	8 (57.14%)	10 (43.47%)	0.156
ARNI	6 (42.85%)	8 (34.78%)	0.129
SGLT2-I	2 (14.28%)	9 (39.13%)	0.068
**Previous NYHA functional class**			
I	0 (0%)	2 (8.69%)	0.684
II	11 (78.57%)	16 (69.6%)	
III	2 (14.28%)	4 (17.4%)	
IV	1 (7.14%)	1 (4.34%)	
**Chronic diuretic treatment**	11 (78.57%)	12 (52.17%)	0.081
**ISGLT2 during follow-up**	7 (50%)	17 (73.9%)	0.131
**Mean follow-up (days +/− SD)**	628.86 +/− 255.028	415.04 +/− 255.08	0.240
**% of time at low risk**	62.17 (+/−21.14)	92.20 (+/−8.267)	**<0.001 ***
**PPV**	0.272 (+/−0.189)	0.0142 (+/−0.053)	**<0.001 ***

ACE-I: Angiotensin-Converting Enzyme Inhibitor; ARA-II: Angiotensin Receptor Antagonist; ARNI: Angiotensin Receptor–Neprilysin Inhibitor; COPD: Chronic Obstructive Pulmonary Disease; CRT-ICD: Cardiac Resynchronization Therapy with Implantable Cardioverter Defibrillator; LVEF: Left Ventricular Ejection Fraction; MRA: Mineralocorticoid Receptor Antagonist; NYHA: New York Heart Association; PPV: Positive predictive value; SGLT2-I: Sodium–Glucose Cotransporter 2 Inhibitor. Percentages represent the number of cases in each category of each of the two groups. * *p* value < 0.05.

## Data Availability

The original contributions presented in this study are included in the article. Further inquiries can be directed to the corresponding author.

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
