# Peer review of "Heart Failure Alert Duration and Time at Risk of Heart Failure as Potential Modifier Factors of the TriageHF Algorithm in Remote Monitoring of Heart Failure: A Cohort Study"

_diagnostics, 2025, doi:10.3390/diagnostics15233065_

Round 1

Reviewer 1 Report

Comments and Suggestions for Authors

Dear editor, 

I reviewed the article entitled “Heart Failure Alerts Duration and Time at Risk of Heart Failure as Potential Modifier Factors of the Triage HF Algorithm in Remote Monitoring of Heart Failure: A Cohort Study”

1- Although the study examines an interesting topic and has the potential to contribute to the literature, Considering the daily clinical attendance of HF patients and the volume of patients implanted with ICDs, the number of patients included in the study was quite insufficient. It is impossible to draw any conclusions with such a small number of patients. The study was underpowered. Neither pre- nor post-hoc power analyses were conducted. Therefore, the study is not powerful enough to reflect clinical practice.

2- What are the main findings? And What is the implication of the main finding? Sections are not understandable. The expression intended to be expressed in the sentences is not understood. There is no coherence of meaning between the sentences.

3-In such an important study testing an algorithm, why was the timeframe chosen so narrow and why were so few patients included? This is extremely restrictive. A large patient population should have been selected.

4- The article should be evaluated in terms of English language. There are some typos in the article.

5- Why did not you perform power analysis?

Comments on the Quality of English Language

Dear editor, 

I reviewed the article entitled “Heart Failure Alerts Duration and Time at Risk of Heart Failure as Potential Modifier Factors of the Triage HF Algorithm in Remote Monitoring of Heart Failure: A Cohort Study”

1- Although the study examines an interesting topic and has the potential to contribute to the literature, Considering the daily clinical attendance of HF patients and the volume of patients implanted with ICDs, the number of patients included in the study was quite insufficient. It is impossible to draw any conclusions with such a small number of patients. The study was underpowered. Neither pre- nor post-hoc power analyses were conducted. Therefore, the study is not powerful enough to reflect clinical practice.

2- What are the main findings? And What is the implication of the main finding? Sections are not understandable. The expression intended to be expressed in the sentences is not understood. There is no coherence of meaning between the sentences.

3-In such an important study testing an algorithm, why was the timeframe chosen so narrow and why were so few patients included? This is extremely restrictive. A large patient population should have been selected.

4- The article should be evaluated in terms of English language. There are some typos in the article.

5- Why did not you perform power analysis?

Author Response

I submit te response to both reviewers.

Response to Reviewer 1

1- Although the study examines an interesting topic and has the potential to contribute to the literature, Considering the daily clinical attendance of HF patients and the volume of patients implanted with ICDs, the number of patients included in the study was quite insufficient. It is impossible to draw any conclusions with such a small number of patients. The study was underpowered. Neither pre- nor post-hoc power analyses were conducted. Therefore, the study is not powerful enough to reflect clinical practice.

Initially there was a larger sample of patients who had had a Medtronic ICD implanted during the period of study, but many of them were ultimately excluded.  There were two main reasons for this phenomenon, the first one is that some of the patients who have had a Medtronic device did not have a device with the Triage-HF function. The second reason is that some of the patients who did have an ICD with the TriageHF function did not have the function activated and therefore had to be excluded. As a result, from our initial sample of 110 patients who have received a Medtronic device, in the end, only 37 of them were suitable for the study. Despite the limited sample of patients, the follow up time resulted in 609 alarms that were evaluated.

Our study had two main objectives; the first one was to explore if modifying the duration threshold could be a potential strategy for the optimization of the TriageHF remote monitoring. The second one was to study if there was a correlation between the time spent in risk during the follow up and the predictive value of the alarms.  For the first objective we carried out a purely descriptive analysis, with no direct statistical comparisons. For the second objective we performed a correlation test which showed that there was a statistically significant relationship between the two variables.

Considering these two aspects we decided not to perform a power analysis, since there was not statistical analysis performed for the first objective and, for the second one, the analysis showed a p value of 0.018, indicating that the study was sufficiently powered to detect statistical differences in this comparison and no type II error was made for the main objectives of the study.

Our conclusion is that, although we conducted a descriptive study with no direct intervention strategy and our sample was limited, what may affect the external validity of our results, the findings are of extreme interest since they open a new non-tested path for trying to make TriageHF remote monitoring more efficient. Further studies should be performed to test these findings.

2- What are the main findings? And What is the implication of the main finding? Sections are not understandable. The expression intended to be expressed in the sentences is not understood. There is no coherence of meaning between the sentences.

The main findings of our study are the following:

  1. Classifying TriageHF moderate risk alarms that do not last for at least 15 days as low risk alarms resulted in an improvement of the specificity and the positive predictive by reducing the false positive alarms, and despite reducing sensitivity, a negative predictive value above 99% was maintained.
  2. Individuals who presented high risk alarms during their follow up were older at the time of implantation, exhibited a higher prevalence of renal disease, experienced a greater frequency of heart-failure episodes, and spent a shorter proportion of time in the low-risk category (Table 3).
  3. There was an inverse correlation between de amount of time spent at risk and the positive predictive value. Thus, those patients who, during follow-up, exhibited a lower proportion of time in the low–heart-failure-risk category demonstrated a higher positive predictive value (R2: 0.64 p: 0.018)

The main implication of these findings is that the TriageHF algorithm can be optimized by disregarding a portion of the moderate-risk alerts and by incorporating the amount of time that patients have spent at a low-risk level during follow-up in the telemonitoring strategies. These findings reflect a simple theorical base, heart failure has a very broad ambulatory spectrum, some patients are clinically stable, and others present continuous exacerbations, and in this second group, alarms will have higher PPV.  According to our results, this approach could reduce the number of false positives requiring evaluation without substantially affecting the negative predictive value.

3-In such an important study testing an algorithm, why was the timeframe chosen so narrow and why were so few patients included? This is extremely restrictive. A large patient population should have been selected.

In the study the mean follow-up was 494 days, with 609 alarms generated. It is true that only 37 patients were finally included; the number of patients included was originally higher, but an important proportion of them were excluded in the end because there were several issues regarding their follow-up by the triageHF algorithm (some of the patients did not have the feature activated even though their ICD had the TriageHF feature).

4- The article should be evaluated in terms of English language. There are some typos in the article.

All the article was revised, and several changes were made.

5- Why did not you perform power analysis?

As it was explained before, since we did not perform a statistical comparison for the first objective, and the second objective was confirmed by a statistically significant correlation, we assumed the study was sufficiently powered to prove our main objectives.

However, it is true that other comparisons were made when dividing the sample according to the level of risk, and some of them showed no statically significant differences between groups. This could be due to an unappreciated type II error which could be ruled out with a power analysis.

Response to Reviewer 2

- The Discussion focuses mainly on algorithmic thresholds and alert duration but could better emphasize how remote monitoring fits into the broader concept of home-based HF management. It would strengthen the manuscript to cite evidence supporting the real-world integration of remote follow-up, telemonitoring, and patient self-management. Suggested addition: After line 272–278 (“Most recent studies are incorporating a hybrid approach, which combines algorithm remote monitorization with a direct phone call…”), recommend inserting: “This perspective aligns with emerging evidence on home-based and telemonitoring approaches for managing heart failure and arrhythmias in patients with cardiac implantable devices, as highlighted by Matteucci et al. (Home Management of Heart Failure and Arrhythmias in Patients with Cardiac Devices during Pandemic. J Clin Med. 2021 Apr 11;10(8):1618. doi: 10.3390/jcm10081618). In that study, remote and home monitoring strategies significantly improved patient outcomes and facilitated early detection of decompensation and arrhythmic events, supporting the integration of telemedicine into standard device follow-up models.”

We followed your recommendation, and we changed the lines that you suggested citing this interesting article.

- While 609 alerts provide a large dataset, only 37 patients were included. The authors should clarify whether statistical power calculations were performed and discuss how the limited population may affect external validity.

As we conducted a descriptive study in which the primary objective did not require a statistical comparison and the regression test for secondary objective showed a p value of < 0.05, we assumed that the study was powered enough to avoid class II errors for the main objectives. However, it is true that even though the p value was < 0,05, the small sample of patients could affect the external validity of our results. It would be interesting to conduct further trials to provide evidence that supports the validity of our findings in a larger population  

- Variables such as medication use (e.g., SGLT2 inhibitors, MRA, ARNI) and comorbidities (CKD, AF) appear unevenly distributed across groups. The statistical section should clarify whether multivariate analyses controlled for these variables to confirm independent associations with PPV or false alerts.

Table 3 shows differences in some variables when dividing the population into three different levels of risk; the first with patients only with low risk, the second one with patients with at least one high risk alarm and the third one with patients with at least one moderate risk alarms but without high-risk alarms.  This table showed differences in several variables, but the amount of false positive alarms in each of the groups were not registered and differences between them were not tested. The correlation comparison between PPV and % in low risk was not carried out using these groups, indeed, it was made considering the variable of time in low risk as a quantitative variable instead of a qualitative one. That’s why the multivariate test was not performed.

-The findings have strong implications for clinical workflow (fewer false positives, reduced staff burden). The authors could quantify the potential reduction in unnecessary interventions.

Even though the study provides interesting data on a potential way of reducing unnecessary interventions we think that our design lacks enough data to try to quantify it. Nevertheless, supported by our results, we suggest combining a strategy of phone calls after a heart failure alarm, but with a variable threshold for the moderate risk alarms, which can vary according to the time spent in low risk by each patient, could be a promising method to reduce unnecessary interventions making this a more efficient strategy.

- Overall language is clear, but some sentences can be streamlined (e.g., “This traduces an HF incidence…” → “This corresponds to an HF incidence…”). Correct consistent spelling (“monitorization” → “monitoring”).

Several changes were made to correct this.

- Table captions should specify whether percentages refer to patients within group or total sample.

- The text uses both “ICD” and “CRT-ICD” interchangeably. Define clearly at first mention and use consistently thereafter.

Patients in our study could have both types of devices, ICD and CRT-ICD, however, 83% of the sample had a CRT-ICD. We added a line to clarify this in the article.

- Several references (e.g., 12, 13, 25, 26) lack full titles or proper formatting. Ensure all include full author lists (up to six authors + et al.), publication year, and DOI.

We corrected all the mistakes in the bibliography and added the DOI in all the references.

- Add the DOI for all ESC Heart Failure references to match MDPI citation style.

We corrected all the mistakes in the bibliography and added the DOI in all the references.

Reviewer 2 Report

Comments and Suggestions for Authors

This is a relevant and well-structured study addressing optimization strategies for the Medtronic TriageHF algorithm in the remote monitoring of heart failure (HF). The topic is timely and contributes to improving the cost-effectiveness and clinical applicability of device-based HF management. The methodology is sound, and the results are clearly presented, though the discussion and contextualization could be deepened in several aspects.

Major Points

- The Discussion focuses mainly on algorithmic thresholds and alert duration but could better emphasize how remote monitoring fits into the broader concept of home-based HF management. It would strengthen the manuscript to cite evidence supporting the real-world integration of remote follow-up, telemonitoring, and patient self-management. Suggested addition: After line 272–278 (“Most recent studies are incorporating a hybrid approach, which combines algorithm remote monitorization with a direct phone call…”), recommend inserting: “This perspective aligns with emerging evidence on home-based and telemonitoring approaches for managing heart failure and arrhythmias in patients with cardiac implantable devices, as highlighted by Matteucci et al. (Home Management of Heart Failure and Arrhythmias in Patients with Cardiac Devices during Pandemic. J Clin Med. 2021 Apr 11;10(8):1618. doi: 10.3390/jcm10081618). In that study, remote and home monitoring strategies significantly improved patient outcomes and facilitated early detection of decompensation and arrhythmic events, supporting the integration of telemedicine into standard device follow-up models.”

- While 609 alerts provide a large dataset, only 37 patients were included. The authors should clarify whether statistical power calculations were performed and discuss how the limited population may affect external validity.

- Variables such as medication use (e.g., SGLT2 inhibitors, MRA, ARNI) and comorbidities (CKD, AF) appear unevenly distributed across groups. The statistical section should clarify whether multivariate analyses controlled for these variables to confirm independent associations with PPV or false alerts.

-The findings have strong implications for clinical workflow (fewer false positives, reduced staff burden). The authors could quantify the potential reduction in unnecessary interventions.

Minor Points

- Overall language is clear, but some sentences can be streamlined (e.g., “This traduces an HF incidence…” → “This corresponds to an HF incidence…”). Correct consistent spelling (“monitorization” → “monitoring”).

- Table captions should specify whether percentages refer to patients within group or total sample.

- The text uses both “ICD” and “CRT-ICD” interchangeably. Define clearly at first mention and use consistently thereafter.

- Several references (e.g., 12, 13, 25, 26) lack full titles or proper formatting. Ensure all include full author lists (up to six authors + et al.), publication year, and DOI.

- Add the DOI for all ESC Heart Failure references to match MDPI citation style.

Comments on the Quality of English Language

see comments

Author Response

Response to Reviewer 1

1- Although the study examines an interesting topic and has the potential to contribute to the literature, Considering the daily clinical attendance of HF patients and the volume of patients implanted with ICDs, the number of patients included in the study was quite insufficient. It is impossible to draw any conclusions with such a small number of patients. The study was underpowered. Neither pre- nor post-hoc power analyses were conducted. Therefore, the study is not powerful enough to reflect clinical practice.

Initially there was a larger sample of patients who had had a Medtronic ICD implanted during the period of study, but many of them were ultimately excluded. There were two main reasons for this phenomenon, the first one is that some of the patients who have had a Medtronic device did not have a device with the Triage-HF function. The second reason is that some of the patients who did have an ICD with the TriageHF function did not have the function activated and therefore had to be excluded. As a result, from our initial sample of 110 patients who have received a Medtronic device, in the end, only 37 of them were suitable for the study. Despite the limited sample of patients, the follow up time resulted in 609 alarms that were evaluated.

Our study had two main objectives; the first one was to explore if modifying the duration threshold could be a potential strategy for the optimization of the TriageHF remote monitoring. The second one was to study if there was a correlation between the time spent in risk during the follow up and the predictive value of the alarms. For the first objective we carried out a purely descriptive analysis, with no direct statistical comparisons. For the second objective we performed a correlation test which showed that there was a statistically significant relationship between the two variables.

Considering these two aspects we decided not to perform a power analysis, since there was not statistical analysis performed for the first objective and, for the second one, the analysis showed a p value of 0.018, indicating that the study was sufficiently powered to detect statistical differences in this comparison and no type II error was made for the main objectives of the study.

Our conclusion is that, although we conducted a descriptive study with no direct intervention strategy and our sample was limited, what may affect the external validity of our results, the findings are of extreme interest since they open a new non-tested path for trying to make TriageHF remote monitoring more efficient. Further studies should be performed to test these findings.

2- What are the main findings? And What is the implication of the main finding? Sections are not understandable. The expression intended to be expressed in the sentences is not understood. There is no coherence of meaning between the sentences.

The main findings of our study are the following:

  1. Classifying TriageHF moderate risk alarms that do not last for at least 15 days as low risk alarms resulted in an improvement of the specificity and the positive predictive by reducing the false positive alarms, and despite reducing sensitivity, a negative predictive value above 99% was maintained.
  2. Individuals who presented high risk alarms during their follow up were older at the time of implantation, exhibited a higher prevalence of renal disease, experienced a greater frequency of heart-failure episodes, and spent a shorter proportion of time in the low-risk category (Table 3).
  3. There was an inverse correlation between de amount of time spent at risk and the positive predictive value. Thus, those patients who, during follow-up, exhibited a lower proportion of time in the low–heart-failure-risk category demonstrated a higher positive predictive value (R2: 0.64 p: 0.018)

The main implication of these findings is that the TriageHF algorithm can be optimized by disregarding a portion of the moderate-risk alerts and by incorporating the amount of time that patients have spent at a low-risk level during follow-up in the telemonitoring strategies. These findings reflect a simple theorical base, heart failure has a very broad ambulatory spectrum, some patients are clinically stable, and others present continuous exacerbations, and in this second group, alarms will have higher PPV. According to our results, this approach could reduce the number of false positives requiring evaluation without substantially affecting the negative predictive value.

3-In such an important study testing an algorithm, why was the timeframe chosen so narrow and why were so few patients included? This is extremely restrictive. A large patient population should have been selected.

In the study the mean follow-up was 494 days, with 609 alarms generated. It is true that only 37 patients were finally included; the number of patients included was originally higher, but an important proportion of them were excluded in the end because there were several issues regarding their follow-up by the triageHF algorithm (some of the patients did not have the feature activated even though their ICD had the TriageHF feature).

4- The article should be evaluated in terms of English language. There are some typos in the article.

All the article was revised, and several changes were made.

5- Why did not you perform power analysis?

As it was explained before, since we did not perform a statistical comparison for the first objective, and the second objective was confirmed by a statistically significant correlation, we assumed the study was sufficiently powered to prove our main objectives.

However, it is true that other comparisons were made when dividing the sample according to the level of risk, and some of them showed no statically significant differences between groups. This could be due to an unappreciated type II error which could be ruled out with a power analysis.

Response to Reviewer 2

The Discussion focuses mainly on algorithmic thresholds and alert duration but could better emphasize how remote monitoring fits into the broader concept of home-based HF management. It would strengthen the manuscript to cite evidence supporting the real-world integration of remote follow-up, telemonitoring, and patient self-management. Suggested addition: After line 272–278 (“Most recent studies are incorporating a hybrid approach, which combines algorithm remote monitorization with a direct phone call…”), recommend inserting: “This perspective aligns with emerging evidence on home-based and telemonitoring approaches for managing heart failure and arrhythmias in patients with cardiac implantable devices, as highlighted by Matteucci et al. (Home Management of Heart Failure and Arrhythmias in Patients with Cardiac Devices during Pandemic. J Clin Med. 2021 Apr 11;10(8):1618. doi: 10.3390/jcm10081618). In that study, remote and home monitoring strategies significantly improved patient outcomes and facilitated early detection of decompensation and arrhythmic events, supporting the integration of telemedicine into standard device follow-up models.”

We followed your recommendation, and we changed the lines that you suggested citing this interesting article.

- While 609 alerts provide a large dataset, only 37 patients were included. The authors should clarify whether statistical power calculations were performed and discuss how the limited population may affect external validity.

As we conducted a descriptive study in which the primary objective did not require a statistical comparison and the regression test for secondary objective showed a p value of < 0.05, we assumed that the study was powered enough to avoid class II errors for the main objectives. However, it is true that even though the p value was < 0,05, the small sample of patients could affect the external validity of our results. It would be interesting to conduct further trials to provide evidence that supports the validity of our findings in a larger population

- Variables such as medication use (e.g., SGLT2 inhibitors, MRA, ARNI) and comorbidities (CKD, AF) appear unevenly distributed across groups. The statistical section should clarify whether multivariate analyses controlled for these variables to confirm independent associations with PPV or false alerts.

Table 3 shows differences in some variables when dividing the population into three different levels of risk; the first with patients only with low risk, the second one with patients with at least one high risk alarm and the third one with patients with at least one moderate risk alarms but without high-risk alarms. This table showed differences in several variables, but the amount of false positive alarms in each of the groups were not registered and differences between them were not tested. The correlation comparison between PPV and % in low risk was not carried out using these groups, indeed, it was made considering the variable of time in low risk as a quantitative variable instead of a qualitative one. That’s why the multivariate test was not performed.

-The findings have strong implications for clinical workflow (fewer false positives, reduced staff burden). The authors could quantify the potential reduction in unnecessary interventions.

Even though the study provides interesting data on a potential way of reducing unnecessary interventions we think that our design lacks enough data to try to quantify it. Nevertheless, supported by our results, we suggest combining a strategy of phone calls after a heart failure alarm, but with a variable threshold for the moderate risk alarms, which can vary according to the time spent in low risk by each patient, could be a promising method to reduce unnecessary interventions making this a more efficient strategy.

- Overall language is clear, but some sentences can be streamlined (e.g., “This traduces an HF incidence…” → “This corresponds to an HF incidence…”). Correct consistent spelling (“monitorization” → “monitoring”).

Several changes were made to correct this.

- Table captions should specify whether percentages refer to patients within group or total sample.

- The text uses both “ICD” and “CRT-ICD” interchangeably. Define clearly at first mention and use consistently thereafter.

Patients in our study could have both types of devices, ICD and CRT-ICD, however, 83% of the sample had a CRT-ICD. We added a line to clarify this in the article.

- Several references (e.g., 12, 13, 25, 26) lack full titles or proper formatting. Ensure all include full author lists (up to six authors + et al.), publication year, and DOI.

We corrected all the mistakes in the bibliography and added the DOI in all the references.

- Add the DOI for all ESC Heart Failure references to match MDPI citation style.

We corrected all the mistakes in the bibliography and added the DOI in all the references.

Round 2

Reviewer 1 Report

Comments and Suggestions for Authors

authors responded in a satisfactory way.

Reviewer 2 Report

Comments and Suggestions for Authors

No other comments